# Reduction in global area burned and wildfire emissions since 1930s enhances carbon uptake by land

Vivek K. Arora[1] & Joe R. Melton [2]

The terrestrial biosphere currently absorbs about 30% of anthropogenic $CO_2$ emissions. This carbon uptake over land results primarily from vegetation's response to increasing atmospheric $CO_2$ but other factors also play a role. Here we show that since the 1930s increasing population densities and cropland area have decreased global area burned, consistent with the charcoal record and recent satellite-based observations. The associated reduced wildfire emissions from increase in cropland area do not enhance carbon uptake since natural vegetation that is spared burning was deforested anyway. However, reduction in fire $CO_2$ emissions due to fire suppression and landscape fragmentation associated with increases in population density is calculated to enhance land carbon uptake by 0.13 Pg C yr$^{-1}$, or ~19% of the global land carbon uptake (0.7 ± 0.6 Pg C yr$^{-1}$), for the 1960–2009 period. These results identify reduction in global wildfire $CO_2$ emissions as yet another mechanism that is currently enhancing carbon uptake over land.

[1] Canadian Centre for Climate Modelling and Analysis, Climate Research Division, Environment and Climate Change Canada, Victoria, BC V8W 2Y2, Canada. [2] Climate Research Division, Environment and Climate Change Canada, Victoria, BC V8W 2Y2, Canada. Correspondence and requests for materials should be addressed to V.K.A. (email: vivek.arora@canada.ca)

It is well known that the atmospheric $CO_2$ concentration ($[CO_2]$) is rising in response to increasing anthropogenic fossil fuel emissions of $CO_2$ and due to land use change (LUC) emissions associated with deforestation of natural vegetation, as a result of increases in cropland and pasture area. The area under crops and agriculture has increased from about 5 million $km^2$ in 1850s to about 15 million $km^2$ for the present day[1] while the world population has increased from about 1 billion to over 7 billion today[2]. It is also well known that only about half of anthropogenic $CO_2$ emitted into the atmosphere stays in the atmosphere while the rest is absorbed by land and ocean[3,4]. For the period 2006–2015, only 45% of the $9.3 \pm 0.5 \, Pg \, C \, yr^{-1}$ anthropogenic emissions are estimated[3] to stay in the atmosphere while the land (30%, $3.1 \pm 0.9 \, Pg \, C \, yr^{-1}$) and ocean (25%, $2.6 \pm 0.5 \, Pg \, C \, yr^{-1}$) took up the rest.

This positive global net flux of carbon from the atmosphere to the land, which indicates a carbon sink, is the result of the response of the terrestrial vegetation to multiple forcings[5,6]. These well-recognized forcings include an increase in $[CO_2]$, anthropogenic LUC and nitrogen deposition, and climate change associated with increasing $[CO_2]$ and other greenhouse gases (GHGs), as well as changes in the concentration of atmospheric aerosols. The increase in $[CO_2]$ is considered to be the primary mechanism responsible for enhanced carbon uptake over land driven by the $CO_2$ fertilization of the terrestrial vegetation[6]. An increase in $[CO_2]$ increases the productivity of $C_3$ vegetation which covers about 82% of the vegetated land surface area[7]. Climate change associated with increase in $[CO_2]$ and other GHGs also contributes to carbon uptake by land. For instance, high-latitude vegetation benefits from warmer temperatures and a longer growing season[8] although in the tropics warmer temperature can potentially reduce vegetation productivity. Nitrogen is one of the primary nutrients which limits plant growth and anthropogenic nitrogen deposition alleviates this limitation[6] contributing to increased carbon uptake although excessive nitrogen deposition can also have detrimental effects on vegetation. Anthropogenic LUC and the accompanying global increase in cropland area, in contrast, turns land into a source of carbon as deforested biomass is burned and as it decomposes over the years following deforestation[9]. The net effect of all forcings depends on the geographical location but is positive globally[3] and hence the net carbon uptake by land. This net carbon uptake results in an increase in the carbon density of vegetation and/or soil carbon pools.

The increase in cropland area and population density across the globe over the historical period has had another effect—they both influence area burned by wildfires and thus affect wildfire $CO_2$ emissions. The increase in cropland area decreases area burned by wildfires. Croplands are typically characterized by lower biomass than forests and they also fragment the landscape both of which affect the spread of fire. In contrast to wildfires, agricultural burning of stubble after harvest is practised to clear up fields for a next crop cycle. Even when agricultural fires are considered together with wildfires, the overall effect of increase in cropland area at the global scale is to decrease area burned[10,11]. Direct anthropogenic influences on wildfires are more complex. Humans caused ignitions lead to accidental, as well as intentional, fires which enhance area burned and fire $CO_2$ emissions above what would be caused by natural lightning-caused ignitions alone. However, anthropogenic suppression of wildfires also decreases area burned and fire related emissions.

Here we quantify the effect of increase in cropland area and population density over the 1850–2014 historical period on wildfires and the resulting impact on carbon uptake by land. We show that over the 1850–2014 historical period the area burned and wildfire $CO_2$ emissions first increase but since the 1930s

increasing population densities and cropland area across the globe have acted to decrease area burned. These results are broadly consistent with sediment-charcoal record[12,13] but specifically with the satellite-based[14,15] observational record for the 1997–2014 period. We use the CLASS-CTEM modeling framework which is based on coupled Canadian Land Surface Scheme (CLASS)[16] and Canadian Terrestrial Ecosystem Model (CTEM)[17]. The CLASS-CTEM framework simulates the physical state of the land surface by modeling soil temperature and liquid and frozen soil moisture contents, as well as the snow cover and depth, all of which respond to changes in meteorology. The biological state of the land surface is represented through vegetation which dynamically responds to changes in meteorology and $[CO_2]$ by changing its height, leaf area index, and rooting depth. The model does not include an explicit representation of the nitrogen cycle and it's coupling to the carbon cycle but does include downregulation of photosynthesis as $CO_2$ increases to emulate nutrient constraints on photosynthesis[18,19]. The CLASS-CTEM framework thus provides fluxes of energy, water and $CO_2$ between the atmosphere and the land. This modeling framework also serves as the land surface component in Canadian Earth System Models (ESMs)[19,20].

Wildfire in CLASS-CTEM is represented based on an approach of intermediate complexity[17,21] which has also been successfully used in other models[22–24]. The approach takes all three aspects of the fire triangle (fuel, moisture, and ignition) into account to determine fire occurrence, area burned, and $CO_2$ emissions. The fire module accounts for natural fires caused by lightning but also anthropogenic fires, which are the result of ignitions caused by humans. In the model, as the population density increases so does the probability of fire ignitions caused by humans. Increasing population density, however, also leads to increased suppression of wildfires. The suppression of wildfires represents fire-fighting efforts, landscape fragmentation and other processes which lead to a reduction in area burned through an increase in the fire extinguishing probability in the model. Both increase in fire ignitions and fire suppression by humans are not explicitly modeled but implicitly expressed as a function of population density. The net result is that in the absence of natural ignition due to lightning the area burned first increases as population density increases but soon enough direct suppression of fire becomes more effective so there is an optimum population density at which the area burned is maximized. However, as lightning increases the primary effect of increasing population density is to reduce area burned. The fire model including the role of population density in anthropogenic fires and fire suppression are described in more detail in the Methods section. The response of global fire behavior to changes in population evolves as geographical changes in population density occur over time. This response is not specified a priori in the model but rather an emergent model behavior depending on geographical changes in population densities. Changes in land cover driven by increases in crop area affect area burned since crop area is assumed not to burn in the model, even if climatic conditions and population density permit otherwise, thus contributing to indirect reduction in area burned by wildfires and fire $CO_2$ emissions[10,25]. The model does not represent agricultural fires since the focus is on the behavior of large scale wildfires and their interaction with the carbon cycle.

We analyze simulations performed over the historical period (1851–2014) that are driven by changes in $[CO_2]$, land use (driven by increases in crop area), population density, and by meteorological data that are based on the CRU-NCEP reanalysis product. The CLASS-CTEM model is run at ~2.8° spatial resolution globally driven with half-hourly meteorological data. Six simulations are performed. The first historical simulation

includes the effect of all forcings. The next four simulations use one forcing at a time while other forcings are kept at their pre-industrial level corresponding to year 1850. The last sixth simulation includes the effect of all forcings, like the first historical simulation, but the geographical distribution of population density is kept at its 1960 value after 1960. The land cover, other forcing data sets and the modeling framework are explained in detail in the Methods section.

## Results

**Changes in area burned and fire $CO_2$ emissions.** Figure 1 shows the time series of global annual area burned (Fig. 1a) and global wildfire $CO_2$ emissions (Fig. 1b) from the historical simulation which includes the effect of all forcings and from the two simulations which include the effect of changes in population density and changes in land use implemented individually. The effects of these two forcings on area burned and fire $CO_2$ emissions are the most dominant. The effect of climate change and increase in $[CO_2]$ on fire behavior is small compared to the effect of changes in population density and in land use. The effect of all forcings on area burned and fire $CO_2$ emissions is shown in Supplementary Fig. 1. The increase in crop area over the historical period contributes to a continuous decrease in area burned and fire $CO_2$ emissions, since crop area is assumed to not burn in the model (orange lines in Fig. 1). This modeled response is consistent with the real world, where permanent agriculture reduces the area potentially burned in natural and managed ecosystems[10,25]. The effects of changes in population density on area burned and fire $CO_2$ emissions are not monotonic (magenta lines) as for the increase in cropland area. Increases in human-caused fire ignitions associated with population density contribute to an increase in area burned, and fire emissions up until about 1950. After 1950, the effect of suppression of wildfires more than

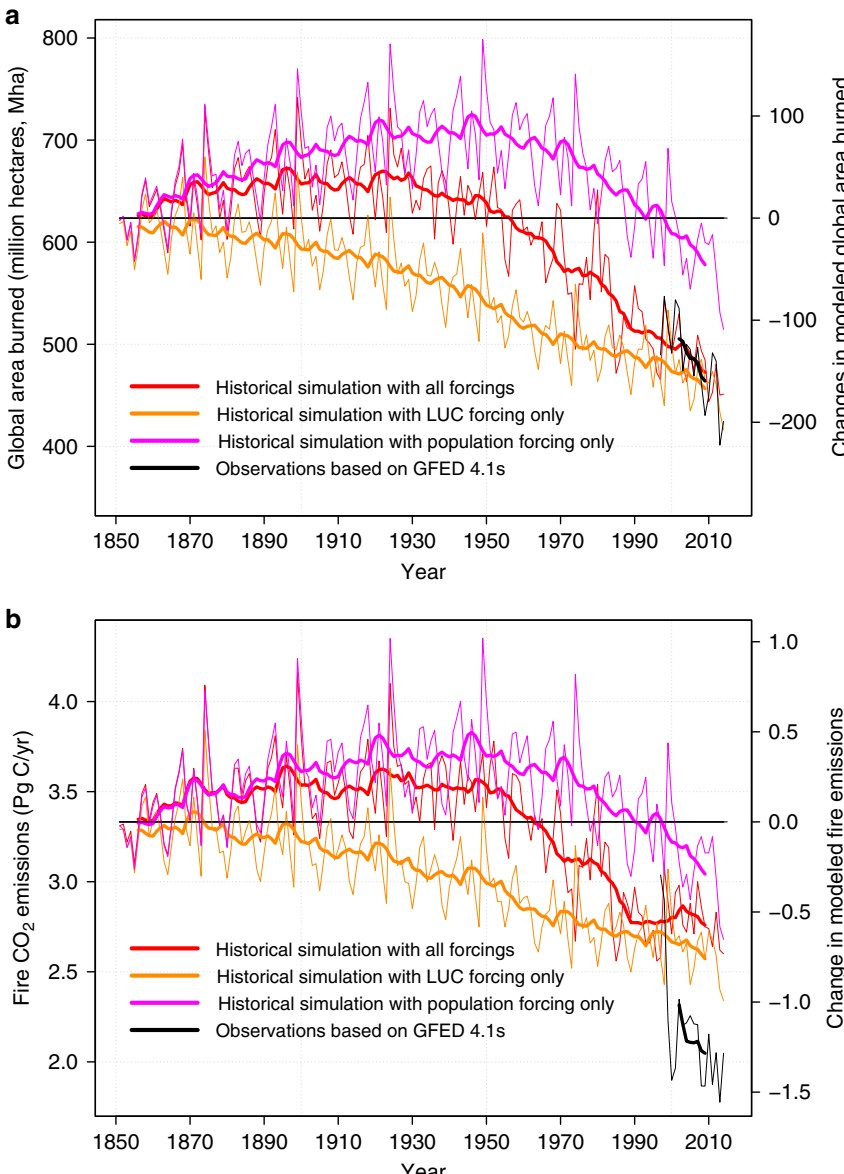

**Fig. 1** Simulated global burned area and fire $CO_2$ emissions. Global area burned (**a**) and fire $CO_2$ emissions (**b**) from the historical simulation driven with all forcings (red line) and driven with population (magenta line) and land use change (LUC) (orange line) forcings individually. The thick lines are the 10-year moving averages. Observation-based burned area (black line) in **a** is based on the GFED4.1s data set and emissions in **b** are based on GFED4.1s burned area, specified combustion factors, and vegetation biomass estimate from the CASA model

compensates for the effect of human-caused fire ignitions so the global area burned, and emissions, start to decrease. The combined effect of all forcings (red lines) is that area burned and emissions increase slightly, compared to their pre-industrial values, up until about 1930 after which they start decreasing. By around 1950s the area burned and emissions reach their pre-industrial levels and continue to decrease thereafter.

**Comparison with observations**. The simulated area burned and fire $CO_2$ emissions are assessed against estimates from the Global Fire Emissions Database[14,15] (GFED, version 4.1 s) in Fig. 1a, b, respectively. GFED estimates provide a 17-year record of satellite-based estimates for the present day (1997–2014) but these estimates only go back to 1997 (see Methods). In Fig. 2, therefore, we also compare simulated area burned with global charcoal indices from the Global Charcoal Database[12,13] (GCD, based on 2008[12] and 2016[13] publications) which allow the assessment of the simulated trends over the full length of our historical simulation, a decadal reconstruction of global burned area from for the 20th century based on land use practices, qualitative reports and local studies from Mouillot and Field[26], and a second satellite-based area burned product[27] from European Space Agency's Climate Change Initiative (ESA CCI) for the period 2005–2011. In Figs. 1a and 2 the simulated global area burned, its trend, and its inter-annual variability over the 1997–2014 period compare well to the GFED4.1s satellite-based estimate. Model and GFED4.1 s average burned area over this period are 483.4 and 485.5 million hectares year$^{-1}$ and their trends are $-5.57 \pm 1.25$ and $-3.43 \pm 1.05$ million hectares year$^{-2}$, respectively (the average area burned over the 2001–2010 period in GFED version 4s[15] is slightly lower at 464.3 million hectares year$^{-1}$). The correlation between simulated and GFED4.1 s annual global area burned estimates is 0.75 for the 1997–2014 period. The area burned in ESA CCI product, in Fig. 2, is lower at 346.2 million hectares year$^{-1}$ and its trend is $-10.38 \pm 5.97$ million hectares year$^{-2}$. The negative trends in the model and both satellite-based products indicate that burned area

has been decreasing. The trends in simulated and satellite-based area burned products are not statistically different. Their $\overline{x} \pm 1.385\sigma_x$ ranges ($\overline{x}$ represents the mean and $\sigma_x$ the standard deviation) overlap implying that the estimates from the two satellite-based sources and the model are statistically not different at the 95% confidence level[28]. The area burned in the ESA CCI product (346.2 million hectares year$^{-1}$) is lower than in the GFED 4.1s product (485.5 million hectares year$^{-1}$) because the former doesn't take into account small fires, while small fires (which include agricultural fires) are taken into account in the GFED 4.1s product.

Winds and smoke carry charcoal from fires and deposit it onto aquatic sediments and therefore sediment-charcoal records provide a proxy for burning. Charcoal records have been shown to reflect trends in biomass and area burned[29,30]. Charcoal records, however, only indicate whether burning is higher or lower relative to a point in time. As a result, the absolute global charcoal index numbers are different between the data based on the 2008 and 2016 releases. Charcoal indices also remain highly uncertain and must be interpreted with caution. Substantial uncertainty exists within each chronology for every contributing sediment record. There is also spatial uncertainty due not only to limited spatial coverage of the records but also due to the varying source areas that each record contributes. In Fig. 2 the overall trend in CLASS-CTEM simulated burned area, for the historical simulation with all forcings, is broadly consistent with the charcoal index based on the 2008[12] publication. The pattern of small increase in area burned and fire emissions from 1851 to about 1930 and the decreasing trend thereafter in the simulation with all forcings in the model is primarily the result of a combination of model response to changes in population density and increasing crop area. This pattern is not reproduced when these forcings are used individually (Fig. 1). The simulated 1851–2014 pattern in model simulated area burned compares much better with the charcoal data released in 2008[12] than that in 2016[13]. The charcoal data released in 2016 also shows a large increase for the 5-year period centered on 2010. This increase in

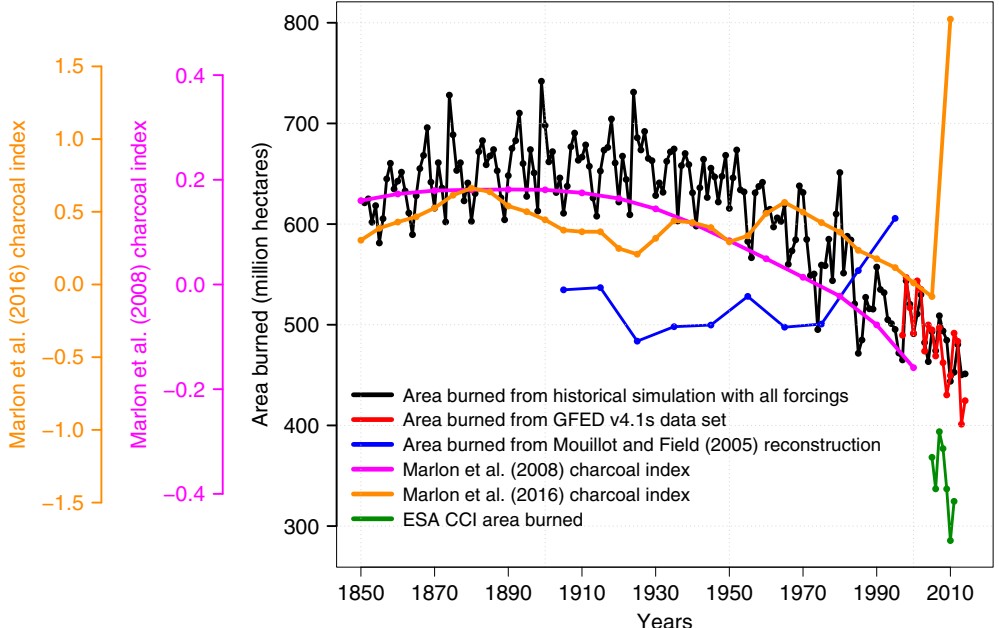

**Fig. 2** Comparison of simulated area burned with other observation-based estimates and charcoal indices. The satellite-based products include the GFED 4.1 s[14,15] data set for the period 1997–2014 and the European Space Agency's Climate Change Initiative (ESA CCI) for the period 2005–2011. Global charcoal indices are from the Global Charcoal Database[12,13] (GCD, based on 2008[12] and 2016[13] publications). The decadal reconstruction of global burned area for the 20th century from Mouillot and Field[26] is based on land use practices, qualitative reports and local studies

fire activity is neither seen in GFED 4.1s area burned or fire $CO_2$ emissions nor in the ESA CCI area burned product. Finally, the decadal reconstruction based on Mouillot and Field[26] shows an increase in area burned during 1990 and 2000 that is not consistent with charcoal records.

In Fig. 1b, averaged over the period 1997–2014, modeled fire $CO_2$ emissions (2.8 Pg C year$^{-1}$) are higher than GFED-based estimates of 2.1 Pg C year$^{-1}$ (but these estimates are themselves based on vegetation biomass estimates from the CASA model, see Methods). Although simulated average fire $CO_2$ emissions are higher than GFED estimates, it is the trend in fire $CO_2$ emissions that impacts the land carbon uptake. The GFED fire $CO_2$ emissions show large values for years 1997 and 1998 associated with peatland fires in Indonesia. Peatland fires are not represented in our modeling framework. However, when the anomalous years 1997 and 1998 are removed, the trend in GFED emissions ($-0.016 \pm 0.009$ Pg C year$^{-2}$) compares well with, and is not statistically different from, the modeled estimates ($-0.014 \pm 0.006$ Pg C year$^{-2}$) for the period 1999–2014. As a result, the decrease in emissions over the 1999–2014 period is statistically similar for both GFED 4.1s and modeled fire $CO_2$ emissions. The contribution of peat fires to other years is around 3%[31] but airport visibility data appears to suggest the frequency of peatland fires in Indonesia is increasing[32]. Peatland fires emit large

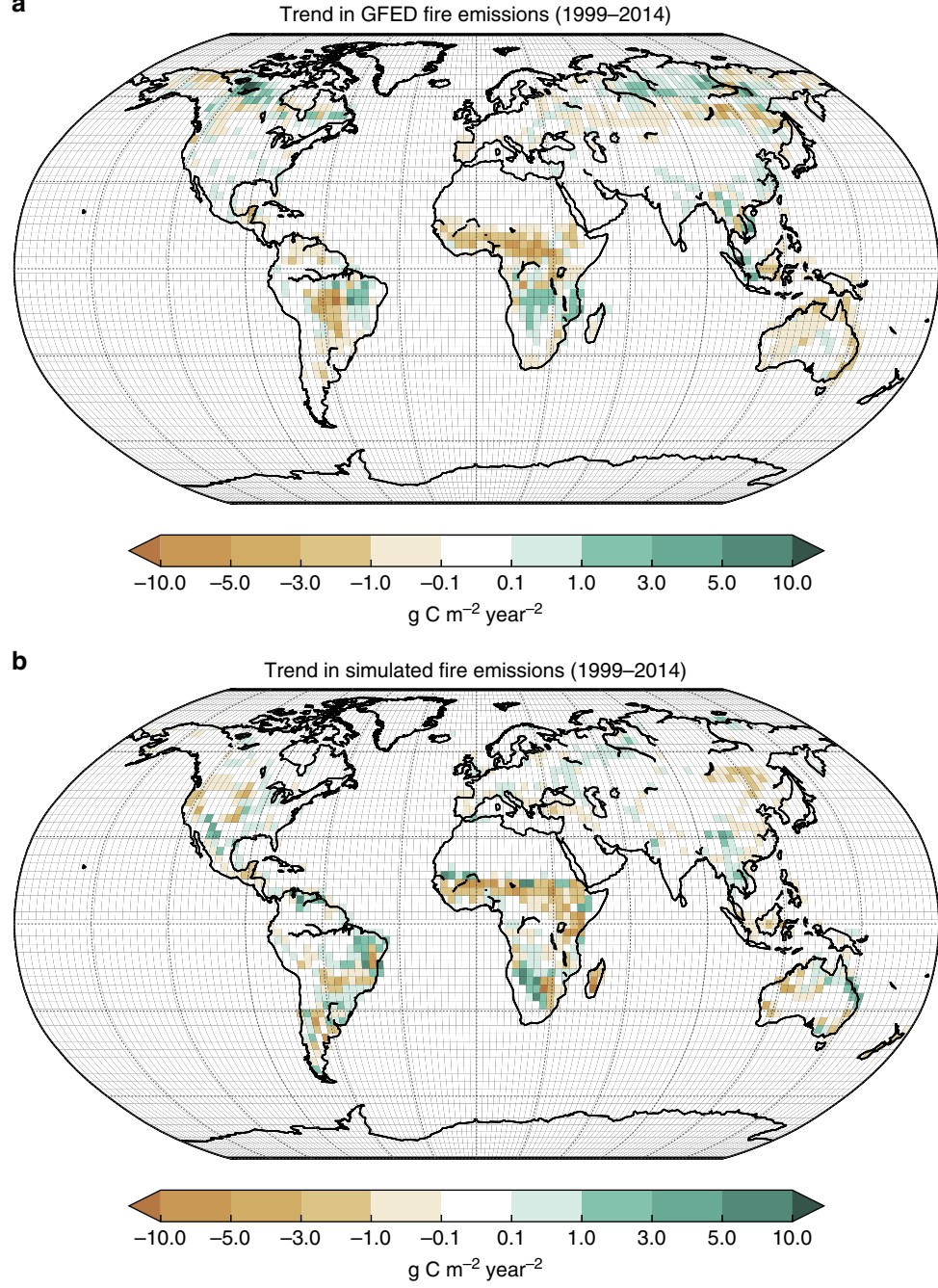

**Fig. 3** Comparison of simulated and observed geographical distribution of trend in fire emissions for the 1999–2014 period. **a** shows the estimates based on fire emissions from the GFED4.1s[14,15] data set and **b** shows the trend based on the results from historical simulation with all forcings

amounts of carbon despite small area burned because of their high carbon density and therefore have the potential to affect the trends in carbon emissions significantly. The trend of decreasing fire emissions over the historical period is also consistent with the decrease in southern hemisphere's atmospheric concentration of CO associated with fire emissions from late 1800s to present day[33].

Figure 3 shows the simulated geographical distribution of trend in fire emissions that compares well with GFED 4.1s estimates especially in the tropical regions that mainly determine the global trend (Fig. 2). This indicates that, at least, in the tropics the modeled fire response to increases in crop area and population densities is reasonably realistic. The model is, however, unable to capture the geographical distribution of trend in fire emissions from high-latitude boreal forests. The GFED4.1s (0.002 Pg C year$^{-2}$) and simulated (0.0002 Pg C year$^{-2}$) trend in fire emissions above 50 °N are, however, an order of magnitude smaller than the trend in global fire emissions (−0.016 Pg C year$^{-2}$). The overall decreasing trend in fire emissions in both the GFED4.1 s data set and historical simulation with all forcings originates from regions south of 50 °N.

**Effect of individual forcings.** Figure 4 shows the effect of individual forcings on fire $CO_2$ emissions for the period 1960–2009 and illustrates the role of increase in population density and crop area in reducing fire $CO_2$ emissions. We choose the 1960–2009 period to allow comparison with Le Quere et al.[3] estimate of net positive atmosphere-land $CO_2$ exchange (i.e., carbon uptake by land) of 0.7 ± 0.6 Pg C yr$^{-1}$. Cumulatively over the 1960–2009 50-year period reduced fire emissions amount to about 19 Pg C which is equivalent to about 0.38 Pg C yr$^{-1}$. Increases in population density and crop area are calculated to yield cumulative reduced fire emissions of 8.9 and 5.7 Pg C, respectively, for the period 1960–2009. These results are based on simulations in which changes in population density and crop area are implemented individually. Fire emissions represent only one component of the net atmosphere-land $CO_2$ flux and not all of the reduction in fire emissions leads to an enhancement of the land carbon sink, as discussed in the next section.

The trend in simulated fire emissions over the period 1960–2009 is −0.016 ± 0.001 Pg C year$^{-2}$, statistically similar to that for the 1999–2014 period. As mentioned above, the trends in simulated and GFED-based fire $CO_2$ emissions for the period 1999–2014 are statistically similar (−0.014 ± 0.006 and −0.016 ± 0.009 Pg C year$^{-2}$, respectively) so a similar cumulative reduction in fire emissions of around 19 Pg C can also be obtained by linearly extrapolating GFED4.1s-based emissions back in time to 1960. In addition, the longer the decreasing trend in fire emissions continues the bigger its impact. A decreasing trend of −0.016 ± 0.009 Pg C year$^{-2}$ over the 15 year period (1999–2014) in GFED data implies that after 15 years the emissions have reduced by 0.24 ± 0.14 Pg C year$^{-1}$. Averaged over the entire duration this means the fire emissions have reduced by half of this amount, i.e., 0.12 ± 0.07 Pg C year$^{-1}$.

## Discussion

The caveat with reduced wildfire fire emissions specifically due to LUC associated with increase in crop area (as seen in Fig. 4) is that this reduction does not enhance the land carbon sink. This is because the vegetation that is spared burning from wildfires was already deforested in the first place. Although croplands can be more productive than natural vegetation they replace, most regions become a source of carbon after conversion to croplands[34] due to decomposition of deforested biomass over years that follow but also due to carbon losses from soil as a result of increased tillage over croplands and reduced carbon inputs into the soil. From the atmospheric budget perspective it does not matter if carbon is emitted from burning of natural vegetation or via deforestation related processes. At the global scale the effect of anthropogenic land use change is to make land a source of carbon in the real world[3] and also in our model. Since the deforestation of natural vegetation and conversion to cropland in the model makes land a much greater source of carbon than the subsequent reduction in fire emissions the reduction in fire emissions does not matter. This is shown in Supplementary Fig. 2 which displays cumulative net atmosphere-land $CO_2$ flux in response to all modeled terrestrial ecosystem processes. In the simulation that is driven with land use change forcing alone although the fire

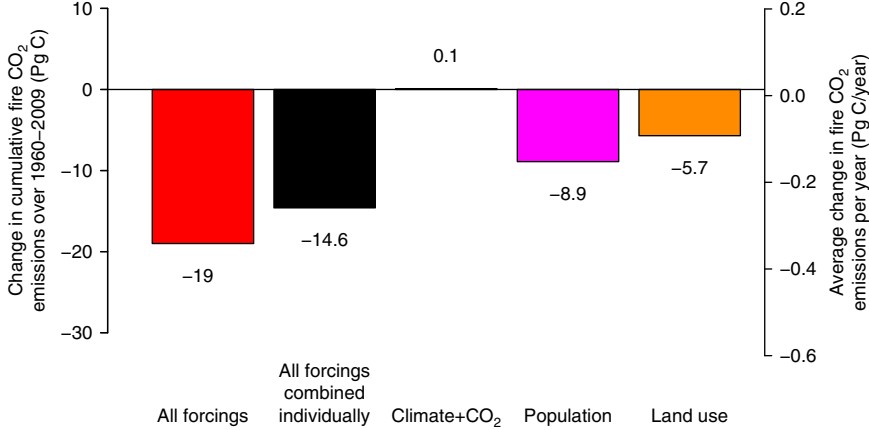

**Fig. 4** Effect of changes in forcings on simulated cumulative fire $CO_2$ emissions over the 1960–2009 period. The secondary y-axis on the right hand side is equal to the cumulative amount divided by 50 (equal to number of years in the 1960–2009 period). For example, the red bar shows the cumulative change in fire emissions from the simulation with all forcings. Since the number is negative it implies that over the 1960–2009 period cumulatively fire emissions decreased by 19 Pg C or equivalently 0.38 Pg C/yr when read on the secondary y-axis. However, reduced fire emissions do not necessarily lead to an enhanced land carbon sink as discussed in the text. The effect of climate and $CO_2$ is combined into a single value since it is very small overall. The individual contributions, calculated from simulations with individual forcings, do not sum together to yield the same decrease in cumulative fire $CO_2$ emissions as the simulation with all forcings because of the spatial correlations between different forcings (e.g. crop area increase correlates positively with population increases) but also the non-linear response of the model to individual forcings

emissions decrease over time (orange line in Fig. 1b) the cumulative net atmosphere-land $CO_2$ flux (orange line in Supplementary Fig. 2) remains negative throughout the simulation indicating that overall land becomes a source of carbon due to land use change.

The reduction in wildfire $CO_2$ emissions from increased direct fire suppression efforts and landscape fragmentation as population density increases, however, does enhance the land carbon sink. We quantify the effect of population changes since 1960 using our last historical simulation in which geographical distribution of population density is kept at 1960 levels after the year 1960 while all other forcings evolve in time. The difference between this and the historical simulation driven with all forcings allows to quantify the impact of population changes since 1960 in a much cleaner manner. Supplementary Fig. 3 illustrates how population change since 1960 contribute to continued reduction in area burned and fire $CO_2$ emissions. This reduction in fire emissions contributes 7.8 Pg C to cumulative reduction in fire emissions over the 1960–2009 period. Indeed, as a result, the net atmosphere-land $CO_2$ flux in the simulation driven with all forcings is higher than in the simulation in which population densities are kept at their 1960 levels after 1960 because continually increasing population densities in the historical simulation with all forcings decrease global fire emissions and thus enhance land carbon uptake (Supplementary Fig. 4). Cumulatively this difference in net atmosphere-land $CO_2$ flux amounts to 6.7 Pg C (similar to cumulative reduced fire emissions of 7.8 Pg C), equivalent to a realized sink of about 0.13 Pg C yr$^{-1}$ or ~19% of the global rate of land carbon uptake (0.7 ± 0.6 Pg yr$^{-1}$) based on the 2016 Global Carbon Project report[3]. As opposed to the reduced fire emissions due to increase in crop area, the reduction in fire emissions associated with population increase thus yields a truly enhanced land carbon uptake.

The statistically similar decreasing trends in simulated and GFED 4.1 s based wildfire emissions are the result of response of the model and the real world fire behavior, respectively, to multiple forcings. Unlike the real world, however, the modeling framework used here allows us to evaluate the response to changes in climate, [$CO_2$], land use and population forcings individually and to quantify their impact in the context of the global rate of land carbon uptake.

In regards to comparison of simulated area burned with observation-based estimates and other proxies (in Fig. 2), the satellite-based GFED4.1s product provides the most reliable estimate. Charcoal records and other reconstructions are proxies with large and unquantified uncertainties. In this regard, model results provide means to look back in time—provided, of course, model performance for the present day and it's response to primary forcings is realistic. We have used trends in global area burned and fire $CO_2$ emissions (Figs. 1 and 2), geographical distribution of trends in area burned (Fig. 3), simulated net atmosphere-land $CO_2$ flux (Supplementary Fig. 4), geographical and zonal distributions of area burned (Supplementary Figs. 5 and 6), and seasonality of simulated global fire $CO_2$ emissions (Supplementary Fig. 6) to illustrate that the model behaves reasonably realistically for the present day. The model response to increase in crop area is straight forwardly interpreted since cropland is not assumed to burn and consistent with existing literature[10,11,25]. However, the model response to increase in population density is not monotonic. This is because increase in fire ignitions and fire suppression by humans are both expressed as a function of population density (see Methods and Supplementary Fig. 7). To illustrate that the model response to population density is realistic we resort to an analysis similar to that of Bistinas et al.[11] who attempt to look for emergent patterns between area burned and its primary drivers. Figure 5 plots

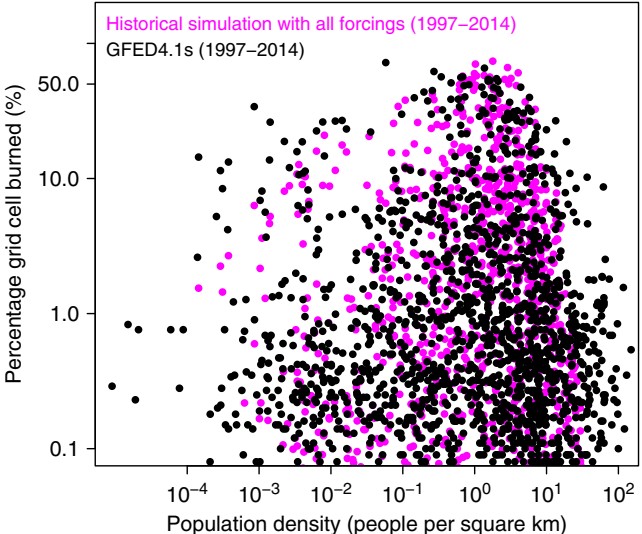

**Fig. 5** The emergent behavior between area burned and population density. Population density and percentage area burned are both plotted on log scale for easy visualization of their behavior at their low values. Comparison is made between simulated (historical simulation with all forcings) and satellite-based (GFED4.1s) values of area burned. The model successfully reproduces the unimodal relationship between population density and area burned which peaks at around 20 people/km$^2$

population density versus percentage area burned (both on log scale) for the GFED4.1s and simulated area burned values averaged over the 1997–2014 period. Each dot in the plot represents a grid cell with non-zero area burned and population density. The model successfully reproduces the unimodal relationship between population density and area burned which peaks at around 20 people km$^{-2}$. The emergent behavior of area burned with respect to population density in the model compares very well to that based on GFED4.1s area burned product and provides confidence in modeled response to population density. As Bistinas et al.[11] explain this unimodal relationship emerges from the fact that population density, net primary productivity (NPP, a measure of carbon uptake by plants), and fire are spatially correlated. People do not live in areas where NPP is low (arid regions where fire is also consequently low) and at high NPP values (in regions with high precipitation) fire is limited by high soil moisture content.

Uncertainty remains in our reported magnitude of the effect of reduction in wildfire on the land carbon sink primarily because of structural model limitations. Our modeling framework simulates only wildfires and we do not take into account deforestation related fires, peatland fires and burning of agricultural stubble after harvest. Agricultural burning of stubble after harvest is a common practice in many parts of the world. Deforestation and agricultural fires, however, typically contribute only ~10–15% to total fire emissions[31] and therefore likely cannot influence the simulated global trend in fire emissions to a large extent. In addition, we calculate the effect of population increase on wildfire emissions and land carbon uptake by differencing two simulations. As such then, the consideration of deforestation fires and burning of agricultural stubble in our modeling framework is not expected to significantly change the calculated magnitude of the effect of a decrease in wildfire emissions on land carbon uptake. The effect of increasing crop area and population density is taken into account in a simple manner in our modeling framework (see Methods). In the real world, anthropogenic fragmentation of the landscape and suppression of fire affect fire regimes in a more

complex manner. Our simple approach is, however, able to capture the observed trends in global area burned and fire $CO_2$ emissions, and the geographical distribution of trends in area burned especially in the tropics.

Satellite-based GFED and ESA CCI estimates of global area burned show a decreasing trend over the short 1997–2014 and 2005–2011 periods, respectively. Our model-based analysis shows that the satellite-based trends of decreasing area burned and fire emissions likely extend back to around the 1930s. The decreasing trend in simulated wildfire $CO_2$ emissions yields reduced fire emissions equivalent to about 0.39 Pg C year$^{-1}$, for the period 1960–2009. All of this reduction, over this period, is attributable to human causes. However, only about 0.13 Pg C year$^{-1}$ of this reduction, that is associated with increasing population densities and overall fire suppression, enhances the current land carbon sink. These results identify reduction in fire $CO_2$ emissions as yet another mechanism that is contributing to the current uptake of carbon by land.

## Methods

**Model set up and simulation design**. The CLASS-CTEM model is driven offline at a spatial resolution of 2.8° which is also the current spatial resolution of Canadian ESMs[19,20]. Version 7 of the CRU-NCEP (Climate Research Unit – National Centre for Environmental Prediction) meteorological data used in this study were available at a spatial resolution of 0.5° and a temporal resolution of 6 h for the 1901–2015 period. A newer version 8 data set of the CRU-NCEP product is now available at https://vesg.ipsl.upmc.fr/thredds/catalog/work/p529viov/cruncep/V8_1901_2016/catalog.html. The CRU-NCEP version 7 data were regridded to a spatial resolution of 2.8° and the data were disaggregated to a half-hourly time step to drive the CLASS-CTEM model[16,17]. The land cover data used to drive the model are based on a geographical reconstruction of the historical land cover driven by the increase in crop area[35] but using the crop area data based on the LUH2 v1h version of the land cover product[1]. The atmospheric $CO_2$ concentration used to drive the model is based on the data provided for use by atmospheric models for the fifth coupled model inter comparison project (CMIP5). The population data are from version 3.2 of the HYDE data set available from ftp://ftp.pbl.nl/../hyde/hyde3.2/baseline/zip/ and based on ref. [2]. The versions of historical simulation with individual and all forcings are initialized from a pre-industrial simulation in which land cover, population and $CO_2$ correspond to the year 1850. In the absence of meteorological data before 1901, the meteorological data for the 1901–1925 period are used repeatedly for the pre-industrial simulation (since the driving meteorological data do not show any significant trends over this period). Once the model reaches equilibrium transient historical simulations are then performed. In simulations in which only the response of LUC, population density changes and $CO_2$ is assessed, the climate forcing is kept at its pre-industrial level by repeatedly using meteorological data for the period 1901–1925 as in the pre-industrial simulation.

**Role of population density**. The fire model is described in more detail in Melton and Arora[17] but its primary features are described here briefly with a particular focus on the role of population density on human-caused fire ignitions and anthropogenic fire suppression. The model calculates probability of fire occurrence ($P_f$) for a representative area ($a_{rep}$) of 500 km$^2$ as

$$P_f = P_b P_m P_i \tag{1}$$

where the right hand side terms represent the fire probabilities that are conditioned on the availability of biomass as a fuel source ($P_b$), the combustibility of the fuel based on its moisture content ($P_m$), and the presence of an ignition source ($P_i$). The rationale for choosing the representative area of 500 km$^2$ is described in Melton and Arora (2016)[17]. $P_b$ varies between 0 and 1 and depends on above-ground biomass so that at biomass density below 0.2 Kg C m$^{-2}$ fire is not sustained and above biomass density of 1.0 Kg C m$^{-2}$ fire is not limited by available biomass. The probability of fire conditioned on the combustibility of the fuel, $P_m$, also varies between 0 and 1 and is dependent on the soil moisture in vegetation's root zone and in the litter layer. The higher the soil moisture the lower the likelihood of fire and lower the value of $P_m$. Finally, the probability of fire conditioned on ignition is calculated as a function of lightning and population density as follows. An initial lightning scalar, $v_F$, that varies between 0 and 1 is found as

$$v_F = \max\left(0, \min\left(1, \frac{F_{c2g} - F_{low}}{F_{high} - F_{low}}\right)\right) \tag{2}$$

where $F_{low}$ and $F_{high}$ represent lower and upper thresholds of cloud-to-ground lightning strikes ($F_{c2g}$, flashes km$^{-2}$ month$^{-1}$), respectively. Below the lower lightning threshold $v_F$ is 0 which implies there are not sufficient lightning strikes to cause fire ignition. Similarly, above the upper lightning threshold $v_F$ is 1 which

implies that there is sufficient lightning so as to not pose a constraint on fire. The lightning scalar, $v_F$, is used to calculate the probability of fire due to natural ignition, $P_{i,n}$, as

$$P_{i,n} = y(v_F) - y(0)(1 - v_F) + v_F[1 - y(1)] \tag{3}$$

$$y(v_F) = \frac{1}{1 + \exp\left(\frac{0.8 - v_F}{0.1}\right)} \tag{4}$$

The value of $P_{i,n}$ gradually increases from 0 to 1 as cloud-to-ground lightning strikes ($F_{c2g}$) increase from $F_{low}$ to $F_{high}$. Fire probability due to ignitions caused by humans, $P_{i,h}$, is parametrized following Kloster et al. (2010)[22] with a dependence on population density, $p_d$ (number of people km$^{-2}$) as

$$P_{i,h} = \min\left[1, \left(\frac{p_d}{300}\right)^{0.43}\right] \tag{5}$$

Finally, the probability of fire conditioned on ignition, $P_i$, is then the total contribution from both natural and human ignition sources

$$P_i = \max\left[0, \min\left\{1, P_{i,n} + (1 - P_{i,n})P_{i,h}\right\}\right] \tag{6}$$

which yields $P_i = P_{i,h}$ in the absence of natural ignition (i.e., when $P_{i,n} = 0$).

The area burned by fires is assumed to be elliptical in shape based on the length to breadth ratio of the ellipse which is a function of wind speed. The specified fire spread rate is dependent on vegetation type but modified as a function of wind speed and soil moisture in the vegetation rooting zone. The fire spread rate determines the length of the elliptical shape of the burnt area, while the length to breadth ratio determines the fire spread rate in the direction perpendicular to the wind speed. With fire spread rate determined, and the geometry of the burned area defined, the fire module then calculates the area burned in 1 day ($a_{1day}$).

The duration of the fire ($\tau$, days) is calculated using the fire extinguishing probability, $q$, which is formulated as

$$q = 0.5 + \frac{\max[0, 1 - \exp(-0.025p_d)]}{2} \tag{7}$$

The fire extinguishing probability, $q$, increases with population density, $p_d$. Equation (7) yields a value of $q$ of 0.5 when $p_d$ is zero and a value of 1 when $p_d$ is infinity. If $q$ represents the probability that a fire will be extinguished on the same day it is initiated then $(1 - q)$ is the probability that it will continue on to the next day. If an assumption is made that individual fire days are independent then $(1 - q)^\tau$ is the probability that on day $\tau$ the fire will still be burning. The probability that a fire will be extinguished on day $\tau$ and thus last exactly $\tau$ days, $P(\tau)$, is thus $q(1 - q)^\tau$. These assumptions yield an exponential distribution of fire duration $\tau$. The expected value of $\tau$ is found as

$$\bar{\tau} = \mathbb{E}(\tau) = \sum_{\tau=0}^{\infty} \tau P(\tau) = \sum_{\tau=0}^{\infty} \tau q(1 - q)^\tau = \frac{1 - q}{q} \tag{8}$$

In the absence of any human influence ($p_d = 0$) $q = 0.5$ (equation 7) and average fire duration $\bar{\tau} = 1$ day. As population density increases, fire suppression increases which increases $q$ and decreases $\tau$ below 1. Based on this fire duration and the area burned in 1 day ($a_{1day}$) the area burned over the duration of the fire ($a_{\tau day}$) is calculated as

$$a_{\tau day} = \mathbb{E}\left(a_{1day}\tau^2\right) = \sum_{\tau=0}^{\infty} a_{1day}\tau^2 q(1 - q)^\tau = a_{1day}\frac{(1 - q)(2 - q)}{q^2}$$
$$\Theta(q) = \frac{(1 - q)(2 - q)}{q^2} \tag{9}$$

since area burned is proportional to $\tau^2$. Finally, area burned for a grid cell with area ($A_g$) is calculated as

$$A_b = P_f a_{\tau day} \frac{A_g}{a_{rep}} = P_f a_{1day} \Theta(q) \frac{A_g}{a_{rep}} \tag{10}$$

Since $P_f = P_b P_m P_i$ this implies area burned is proportional to $P_i \Theta(q)$.

In this framework, population density ($p_d$) increases the probability of fire ignition (equations 5 and 6) and decreases its duration and area burned through fire suppression (equations 7 through 9). The net result of these competing processes is that in the absence of natural ignition due to lightning ($P_{i,n} = 0$) the area burned first increases as population density increases but soon enough direct suppression of fire becomes more effective so there is an optimum population density at which the area burned is maximized. This is shown in Supplementary Fig. 7a. However, this behavior changes as ignition caused by lightning increases

and $P_{i,n}$ becomes positive. As $P_{i,n}$ increases above zero the area burned scalar, $A = P_t\Theta(q)$, basically decreases as population density increases (see Supplementary Fig. 7b, c and d).

**Observation-based data sets.** The simulated burned area and emissions are compared against satellite-based estimates from the GFED4.1s and ESA CCI products and global charcoal indices from the GCD versions based on 2008[12] and 2016[13] publications. The GFED4.1s area burned data are based on combination of MODIS burned area maps with active fire data from the Tropical Rainfall Measuring Mission (TRMM) Visible and Infrared Scanner (VIRS) and the Along-Track Scanning Radiometer (ATSR) family of sensors. The data also includes area burned from small fires[15]. Burned area in the ESA CCI product is based on a hybrid approach, combining information on active fires from the MODIS sensor and temporal changes in reflectance from the MERIS time series[27]. The fire emissions in the GFED data set are calculated using these satellite-based area burned estimates, specified combustion factors and vegetation biomass estimates which are based on the Carnegie Ames Stanford Approach (CASA) model. The 2008 and 2016 versions of the GCD data set are based on 406 and 736 sediment-charcoal records, respectively, from around the globe, most of which are recovered from natural lakes and available as decadal and pentadal data sets, respectively. The other charcoal records are recovered from wetlands but also from soils and from coastal or marine environments.

Not all models reproduce the recent decreasing trend in global area burned. In a recent fire model intercomparison project (FireMIP[36]) that contributed results to Andela et al.[10] (their Fig. 3) five of the participating models (including CLASS-CTEM) simulated a decrease in global area burned over the 1997–2014 period while the remaining four simulated an increase over the same period. The version of the CLASS-CTEM model used here is slightly different than that for FireMIP and yields a somewhat larger negative trend than its results shown in Andela et al.[10].

**Model evaluation.** Supplementary Fig. 4 shows the simulated net atmosphere-land $CO_2$ flux (which is the result of interaction of all process–including photosynthesis, respiration, land use change and fire) from the historical simulation with all forcings and the simulation in which population density is kept at 1960 levels after 1960. These results are compared against observation-based estimates from the Global Carbon Project[3] and show that the model realistically simulates the land carbon sink from 1960s onwards. Simulated fluxes lie within the uncertainty range of observation-based estimates except during 1970s.

Supplementary Fig. 5 compares the geographical distribution of simulated area burned for the period 1997–2014 with the GFED4.1s (1997–2014) and ESA CCI (2005–2011) estimates. Supplementary Fig. 6 compares the simulated latitudinal distribution of area burned (panel a) and seasonality of simulated global fire $CO_2$ emissions (Supplementary Fig. 6b) with GFED-based estimates. In Supplementary Figs. 5 and 6a the model captures the broad spatial pattern of higher area burned in the seasonally-dry tropical regions compared to mid- and high-latitude regions. The area burned in GFED 4.1s data set is higher than in version 3 because it takes into account small fires as well. The model is, however, unable to realistically model area burned in regions which burns less than 0.5% per year but, of course, these areas contribute to a very small fraction to global area burned. Large areas in Eurasia where the model does not capture these small area burned fractions is where croplands exists and likely due to no representation of agricultural fires. In Supplementary Fig. 6b, the seasonality of fire emissions depends on the seasonality in area burned but also the simulated vegetation biomass and its seasonality. The seasonality of fire emissions in the CLASS-CTEM model, of course, depends on its simulated vegetation biomass. The magnitude and seasonality in fire emissions from GFED4.1s and 3 data sets depends on the seasonality in satellite-based area burned but also the magnitude and seasonality of vegetation biomass simulated by the CASA model, from which the GFED fire $CO_2$ emissions are calculated. Overall, CLASS-CTEM is able to capture the broad peak in emissions during August and September.

Supplementary Fig. 8 shows the effect of individual forcings on fire $CO_2$ emissions for the full 1851–2014 period. The effect of changes in population density and land cover change depends on the date since which their effect is calculated. This is the reason for the differences in Fig. 4 and Supplementary Fig. 8. Coincidentally, the effect of changes in all forcings over the full length of the 1850–2014 historical period is very small. Over this period, the increase in fire $CO_2$ emissions caused by changes in climate, $CO_2$ and population is compensated by the decrease caused due to land cover changes. There is very little effect of increasing $CO_2$ on fire emissions in Fig. 4 and Supplementary Fig. 8 and that's the reason for combining its effect with climate. This is because emissions depend on above-ground vegetation biomass and not directly on gross primary productivity (GPP). Vegetation biomass only increases incrementally from one year to next as GPP increases with increasing $CO_2$. Almost 90% of the wildfire emissions are generated in tropical and sub-tropical regions which burn every 2–4 years in the model, and a large fraction of this comes from herbaceous vegetation. The increase in vegetation biomass (due to an increase in $CO_2$) between fire events is not sufficiently large for increasing $CO_2$ to make a significant effect on wildfire emissions over the historical period.

**Code availability.** The model code is available at https://gitlab.com/jormelton/classctem but requires registration on gitlab.com.

**Data availability.** The raw data in NetCDF gridded format for area burned, fire $CO_2$ emissions and other primary terrestrial carbon pools and fluxes for the simulations reported in this paper can be obtained from the first author (vivek.arora@canada.ca).

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

## Acknowledgements
We thank Werner Kurz, Daniel Thompson, John Fyfe, Pierre Friedlingstein, Bill Merryfield and Laxmi Sushama for providing helpful comments on an earlier version of this paper. We specifically want to thank Jennifer Marlon for sharing with us the global charcoal indices based on her 2008 and 2016 publications and her insights into the uncertainty associated with charcoal data. We also thank the GFED team for making their data publicly available, and Nicolas Viovy for making the CRU-NCEP data set available.

## Author contributions
V.A. performed the experiments and wrote the majority of the manuscript. JM wrote parts of the manuscript, performed review of the relevant literature and set up the modeling framework to run CLASS-CTEM model.

## Additional information

**Competing interests:** The authors declare no competing interests.

