## [Peer Review File · Nature Communications]

Reviewers' comments:

Reviewer #1 (Remarks to the Author):

This manuscript argues that since about 1950 there's been a global reduction in the area burned annually and in the emissions of carbon from fires. The results are consistent with satellite (GFED) and charcoal data. The authors attribute the reduced burning to agricultural expansion and an increase in population density, noting that before 1950 increases in population density worked to increase the area burned (not decrease it). Changes in climate and CO₂ have had no effect on the changes. The reduction in carbon emissions since 1950 has been equivalent to 0.39 PgC yr⁻¹. "However, only about 0.13 PgC yr⁻¹ of this reduction, that is associated with increasing population densities and overall fire suppression, is expected to enhance the current land carbon sink." (lines 263-265).

I don't understand this last statement. Why don't the all of the reduced emissions (0.39 PgC yr⁻¹) result in an apparent increase in the carbon sink on land? And what can the other 2/3 of the reduced emissions be attributed to, if not land use, population density, climate, or CO₂? I can understand that deforestation for agricultural lands results in carbon emissions each year, and that associated with this increase in managed lands is a reduction in agricultural burning. But why isn't the entire reduction in emissions (from reduced burning) contributing to the (apparent) uptake of carbon by land (i.e., larger sink), even if only a third is "attributable solely to human causes"? Do the authors agree? Is this a minor point, or have I missed an important distinction?

Specific points

Lines 129-130: "It is well known that permanent agriculture reduces the area potentially burned in natural and managed ecosystems worldwide (Ref)." The reference is missing. I can suggest the following reference, although it applies to the U.S. and not the world:

Houghton, R.A., J.L. Hackler, and K.T. Lawrence. 2000. Changes in terrestrial carbon storage in the United States. 2. The role of fire and fire management. *Global Ecology and Biogeography* 9:145-170.

Reviewer #2 (Remarks to the Author):

This manuscript reports on a simulation study that seeks to partition the influences of different factors on global area burned and emissions of CO₂ from fire. While certainly a worthy task, there

are some issues with the research that are problematic for the results. The two primary ones are that the manuscript suggests the model response to different factors (e.g. population and LUC) was predetermined. Further, there are inconsistencies between the GFED and GCD traces in figure 1 and what is reported in the original citations. Furthermore, I think the manuscript would benefit from focusing on the entire simulation period, rather than 1960-2009 period. I understand that this particular period was identified to facilitate comparison with previous work, but the focus of the study was on a longer period and this should be the focus of the abstract.

L93-99: This section states that the fire module has some predetermined response whereby the influence of population growth on area burned is specified in the model. If the fire module is parameterized such that increasing population density leads to increased ignitions and increased suppression, then it would be impossible to achieve a different result.

Marlon et al. 2016 report that GCDv3 data show a global decline in biomass burning until the Little Ice Age, followed by a gradual increase until the 19th century, followed by a rapid increase in the 20th century. Furthermore, Marlon et al. (2016) report that the documented large decline in biomass burning in the 20th century reported in Marlon et al. (2008) had large uncertainties and their 2016 paper includes new data "with a finer-scale temporal focus". GFED4 mean annual area burned globally is 349.7 Mha from 2000-2012 (Giglio et al. 2013). Figure 1 shows Marlon et al. (2008), which should be updated with the 2016 GCDv3 data. Furthermore, Fig 1 shows GFED4 data and area burned by small fires from Randerson et al. (2012). Randerson et al. (2012) report that the mean burned area over the period 2001-2010, inclusive of small fires was 464.3 Mha/yr. There appears to be some inconsistencies between global area burned in Fig 1 using observations based from GFED4 and values reported in Giglio et al and Randerson et al.

Reviewer #3 (Remarks to the Author):

The finding that the reduction of wildfire emissions substantially increased the carbon uptake by land is a very interesting and important finding. The methods used are appropriate, the model is a state of the art model and the assumptions of the fire model are well supported by literature. Fire has recently been considered rather carbon neutral, as vegetation can grow back after fire and the carbon stocks are refilled rather fast. This study shows that under global change and changing fire regime, fire is an important determinant of the land carbon sink. This finding is novel and will be interesting for the whole carbon cycle community.

I have several comments which i hope will strengthen the manuscript.

Specific comments:

Title: the title sounds rather trivial (reduced emissions increase the land uptake), a reference to the time period might help to make it more specific.

l.16-18: Do you actually show that the croplands do not enhance carbon uptake? Figure 2 suggests the opposite. You might want to indicate the effect of forest removal there.

l. 60: please be clear where you are talking about reality and where about your model
implementation: croplands are not allowed to burn is certainly a model assumption, while most of the other parts of that paragraph seem to describe reality.

l. 68 ff.: by using a model.

l. 70 ff: The behaviour does not look that complex, there is first a slight increase and then a decrease. Do you mean the underlying processes?

l. 75 ff: is nitrogen included, as the process was introduced as being important for the land sink?

l. 95 ff: this is misleading: fire-fighting and landscape fragmentation are not explicitly represented in the model.

l. 101,102: add support from data analysis literature on the reduction of burned area with increasing cropland fraction.

l.106-114: also introduce the simulation with population density fixed at the value of 1960.

l. 110: typo: "meteorological"

l.129: please add the reference!

l.157: as the emissions in total are a lot higher in the model than in the GFED data, it might be better to compare the relative decrease instead of the absolute decrease.

l. 160: The importance of peatland fires should be acknowledged better though. It is a potentially large source of very old carbon that gets emitted to the atmosphere. Moreover, visibility data of airport stations indicate that peatland fires have increased over the last decades (Field and van der Werf, 2009, Nature Geoscience). Boreal peatlands are as far as I know not accounted for in GFED or any modelling approach, but potentially an increasing net source of carbon (as old carbon is emitted).

l.186: I am not sure how this 19PgC compare to the Le Quere estimate. If fire emissions are reduced, then there is more biomass on land leading to higher respiration rates (due to the higher carbon stock). Are the 19Pg C a change in the carbon stocks on land due to the reduced emissions or simply the sum of reduced emissions?

I.203: the decrease in burned area with increasing cropland fraction is usually assumed to be also due to increased landscape fragmentation this would enhance the land carbon sink. Not all croplands were previously forests, moreover croplands often have a higher productivity.

I. 225-228: again is this just based on the reduction of fire emissions or does this actually represent a change in the land carbon uptake due to changes in fire regimes?

I.243: and peatfires.

I. 255: The comparison with the observations however shows that the simple representation is able to capture the global trends in fire emissions.

I.273: What is the family of Canadian ESMs? any reference? Or delete this information, it does not seem important to know.

I.289: typo: meteorological

I.264-265: I don't understand this. With your factorial simulations you can quantify how the land sink changes.

I. 475: ref 28 is not mentioned in the text.

Figure 2: explain what the numbers in the graph are.

Figure S6: you could add the GFAS dataset for carbon emission seasonality. It might also be interesting to add a trend analysis of the GFAS emissions to the paper, although the period is shorter for GFAS.

Figure S7: Where does tau go to?

Response to reviewers' comments for "Reduction in global area burned and wildfire emissions enhances carbon uptake by land"

We thank the three reviewers who reviewed our paper for taking the time to review our manuscript and for their constructive comments. We have addressed all reviewers' comments as indicated below in revising our manuscript and used the generous 5000 word limit to clarify the issues raised by our reviewers. Our responses are indicated below in bold font face.

Reviewer #1 (Remarks to the Author):

This manuscript argues that since about 1950 there's been a global reduction in the area burned annually and in the emissions of carbon from fires. The results are consistent with satellite (GFED) and charcoal data. The authors attribute the reduced burning to agricultural expansion and an increase in population density, noting that before 1950 increases in population density worked to increase the area burned (not decrease it). Changes in climate and CO₂ have had no effect on the changes. The reduction in carbon emissions since 1950 has been equivalent to 0.39 PgC yr⁻¹. "However, only about 0.13 PgC yr⁻¹ of this reduction, that is associated with increasing population densities and overall fire suppression, is expected to enhance the current land carbon sink." (lines 263-265).

I don't understand this last statement. Why don't the all of the reduced emissions (0.39 PgC yr⁻¹) result in an apparent increase in the carbon sink on land? And what can the other 2/3 of the reduced emissions be attributed to, if not land use, population density, climate, or CO₂? I can understand that deforestation for agricultural lands results in carbon emissions each year, and that associated with this increase in managed lands is a reduction in agricultural burning. But why isn't the entire reduction in emissions (from reduced burning) contributing to the (apparent) uptake of carbon by land (i.e., larger sink), even if only a third is "attributable solely to human causes"? Do the authors agree? Is this a minor point, or have I missed an important distinction?

It appears reviewer #1 has somewhat misinterpreted our results. The increase in cropland area in the model (and in the real world) comes from a decrease in the area of natural vegetation. This is deforestation in a general sense and implies clearing of land previously occupied by natural vegetation. As some of the deforested biomass is burned and the rest decomposes over years that follow and soils lose carbon due to increase tillage - overall the land loses carbon creating land use change (LUC) emissions. The LUC emissions are a much greater source of CO₂ to the atmosphere than the fire CO₂ emissions which are now avoided because natural vegetation is no longer there and croplands are not allowed to burn. As a result, the savings (in carbon) from reduced fire emissions which are smaller than LUC emissions do not matter and do not lead to a carbon sink. We have tried to make this point clear in the revised manuscript at several places. Reviewer #1 also misinterpreted that only

0.13 Pg C/yr out of the total of 0.39 Pg C/yr of reduced fire emissions are attributable to human causes. In fact, what the manuscript tried to say is that only 0.13 out of 0.39 Pg C/yr enhances the land carbon sink. We have made changes in the manuscript to clarify this as well.

Specific points

Lines 129-130: "It is well known that permanent agriculture reduces the area potentially burned in natural and managed ecosystems worldwide (Ref)." The reference is missing. I can suggest the following reference, although it applies to the U.S. and not the world:

Houghton, R.A., J.L. Hackler, and K.T. Lawrence. 2000. Changes in terrestrial carbon storage in the United States. 2. The role of fire and fire management. *Global Ecology and Biogeography* 9:145-170.

Thank you for pointing this reference which we now use in our manuscript.

Reviewer #2 (Remarks to the Author):

This manuscript reports on a simulation study that seeks to partition the influences of different factors on global area burned and emissions of CO₂ from fire. While certainly a worthy task, there are some issues with the research that are problematic for the results. The two primary ones are that the manuscript suggests the model response to different factors (e.g. population and LUC) was predetermined. Further, there are inconsistencies between the GFED and GCD traces in figure 1 and what is reported in the original citations. Furthermore, I think the manuscript would benefit from focusing on the entire simulation period, rather than 1960-2009 period. I understand that this particular period was identified to facilitate comparison with previous work, but the focus of the study was on a longer period and this should be the focus of the abstract.

L93-99: This section states that the fire module has some predetermined response whereby the influence of population growth on area burned is specified in the model. If the fire module is parameterized such that increasing population density leads to increased ignitions and increased suppression, then it would be impossible to achieve a different result.

While both human caused fire ignitions and suppression are modelled as a function of population density, the response of global fire behaviour is not specified as a priori but is rather an emergent model behaviour. This response evolves as geographical changes in population density occur over time. Consider the modelling of photosynthesis as a function of CO₂ for C₃ plants which saturates over times, as an analogy. Yes, as CO₂ increases photosynthesis will increase (assuming nutrients are not limiting) but the rate of change of photosynthesis per unit change in CO₂ depends on the initial CO₂. In the case of photosynthesis, the photosynthesis monotonically increases but at a gradually slower rate as CO₂ increases. The response of fire behaviour to increases in population is more complex

than that. Area burned first increases and then decreases as population density increases at a given point so the globally averaged fire response cannot be specified a priori. We have revised our manuscript to make this aspect more clear.

Marlon et al. 2016 report that GCDv3 data show a global decline in biomass burning until the Little Ice Age, followed by a gradual increase until the 19th century, followed by a rapid increase in the 20th century. Furthermore, Marlon et al. (2016) report that the documented large decline in biomass burning in the 20th century reported in Marlon et al. (2008) had large uncertainties and their 2016 paper includes new data “with a finer-scale temporal focus”. GFED4 mean annual area burned globally is 349.7 Mha from 2000-2012 (Giglio et al. 2013). Figure 1 shows Marlon et al. (2008), which should be updated with the 2016 GCDv3 data. Furthermore, Fig 1 shows GFED4 data and area burned by small fires from Randerson et al. (2012). Randerson et al. (2012) report that the mean burned area over the period 2001-2010, inclusive of small fires was 464.3 Mha/yr. There appears to be some inconsistencies between global area burned in Fig 1 using observations based from GFED4 and values reported in Giglio et al and Randerson et al.

We have obtained the newer 2016 charcoal index data from Jennifer Marlon corresponding to her Marlon et al. (2016) paper and we compare our simulated burn area and fire CO₂ emissions against these data in an additional figure in the supplementary information. The reason we do not include these data in the main manuscript is because these data show a large increase in global charcoal index for the 5-year period corresponding to 2010 that is not consistent with the satellite-based global area burned based on the GFED data which shows a decreasing trend. In addition since the absolute values of charcoal indices released in 2008 and 2016 are different it is difficult to plot both charcoal indices on the same figure along with area burned or fire emissions. Such a figure would need two secondary y-axes as opposed to one secondary y-axis as we currently have. We also include additional discussion of uncertainty associated with the charcoal data (following our email exchanges with Jennifer Marlon) to stress that charcoal is neither area burned nor fire CO₂ emissions and that its interpretation must be cautious and conservative.

In regards to inconsistency with the Randerson et al. (2012) number for mean area burned over the 2001-2010 period of 464.3 Mha/yr and our number reported in our manuscript of 485.5 Mha/yr we have checked our scripts and downloaded data again from GFED’s web site. It turns out that the data reported in Randerson et al. (2012) were based on GFED version 4s while the data we downloaded and reported in our manuscript correspond to GFED version 4.1s. We have also cross checked our number with Jim Randerson and he confirmed that the newer data set has slightly higher burned area. Thank you for noting this inconsistency. We have clarified this in revising our manuscript.

Reviewer #3 (Remarks to the Author):

The finding that the reduction of wildfire emissions substantially increased the carbon uptake by land is a very interesting and important finding. The methods used are appropriate, the model is a state of the art model and the assumptions of the fire model are well supported by literature. Fire has recently been considered rather carbon neutral, as vegetation can grow back after fire and the carbon stocks are refilled rather fast. This study shows that under global change and changing fire regime, fire is an important determinant of the land carbon sink. This finding is novel and will be interesting for the whole carbon cycle community.

I have several comments which i hope will strengthen the manuscript.

Specific comments:

Title: the title sounds rather trivial (reduced emissions increase the land uptake), a reference to the time period might help to make it more specific.

Thank you for your suggestion we have added “since 1930s” to our manuscript’s title.

l.16-18: Do you actually show that the croplands do not enhance carbon uptake? Figure 2 suggests the opposite. You might want to indicate the effect of forest removal there.

We have added text at the end of this sentence to make it clear that deforested vegetation releases carbon. In addition, we have added text in the discussion section and an additional figure in the supplementary information to clarify why reduced fire emissions from increase in cropland area do not enhance the land carbon sink despite decrease in fire emissions, as explained in the answer to reviewer #1’s comment.

l. 60: please be clear where you are talking about reality and where about your model implementation: croplands are not allowed to burn is certainly a model assumption, while most of the other parts of that paragraph seem to describe reality.

Thank you for pointing this. We have revised the sentence and the surrounding text to make it clear that this paragraph describe reality.

l. 68 ff.: by using a model.

We have expanded on the text here to imply that this paragraph describes reality.

l. 70 ff: The behaviour does not look that complex, there is first a slight increase and then a decrease. Do you mean the underlying processes?

We have reworded this sentence to mention the increase in area burned early on in the 1851-2014 historical period and the decrease since the 1930s.

l. 75 ff: is nitrogen included, as the process was introduced as being important for the land sink?

The model does not include an explicit representation of the nitrogen cycle and it's coupling to the carbon cycle but does include downregulation of photosynthesis as CO₂ increases to emulate nutrient constraints on photosynthesis. We now mention this in the manuscript.

l. 95 ff: this is misleading: fire-fighting and landscape fragmentation are not explicitly represented in the model.

We have revised the text around this sentence which now clearly says that - both increase in fire ignitions and fire suppression by humans are not explicitly modelled but implicitly expressed as a function of population density.

l. 101,102: add support from data analysis literature on the reduction of burned area with increasing cropland fraction.

We now reference two papers at the end of this sentence to support model assumption of not letting the crop area burn.

l.106-114: also introduce the simulation with population density fixed at the value of 1960.

All simulations are now introduced at the end of the introductory section.

l. 110: typo: "meteorological"

Thank you.

l.129: please add the reference!

Done.

l.157: as the emissions in total are a lot higher in the model than in the GFED data, it might be better to compare the relative decrease instead of the absolute decrease.

We would have done this if we were comparing emissions only over the 1999-2014 GFED period. However, since we show modelled emissions over the full 1851-2014 it would be

difficult to do this. However, the manuscript does mention that it is the trend in emissions that matters and the modelled and GFED fire CO₂ emissions show trends which are statistically similar.

I. 160: The importance of peatland fires should be acknowledged better though. It is a potentially large source of very old carbon that gets emitted to the atmosphere. Moreover, visibility data of airport stations indicate that peatland fires have increased over the last decades (Field and van der Werf, 2009, Nature Geoscience). Boreal peatlands are as far as I know not accounted for in GFED or any modelling approach, but potentially an increasing net source of carbon (as old carbon is emitted).

We now discuss the Indonesian peatlands in context of Field and van der Werf (2009) paper and have also added an additional sentence that increasing fire emissions from peatlands has the potential to affect the trend in fire emissions significantly.

I.186: I am not sure how this 19PgC compare to the Le Quere estimate. If fire emissions are reduced, then there is more biomass on land leading to higher respiration rates (due to the higher carbon stock). Are the 19Pg C a change in the carbon stocks on land due to the reduced emissions or simply the sum of reduced emissions?

We have done two things to address this comment. First, we mention upfront in the introductory section when discussing positive land carbon uptake that net carbon uptake by land results in an increase in the carbon density of vegetation and/or soil carbon pools. Second, after the text where this 19 Pg C of cumulative reduced fire emissions are mentioned we now also mention that fire emissions represent only one component of the net atmosphere-land CO₂ flux and not all of the reduction in fire emissions leads to an enhancement of the land carbon sink.

I.203: the decrease in burned area with increasing cropland fraction is usually assumed to be also due to increased landscape fragmentation this would enhance the land carbon sink. Not all croplands were previously forests, moreover croplands often have a higher productivity.

We now mention in our revised manuscript that croplands can be more productive than the natural vegetation they replace but globally land use change has caused the land to become a source of carbon that is land use emissions are positive globally. We now also mention upfront in our manuscript that one reason croplands reduce burned area is due to landscape fragmentation (line 91 of the revised manuscript).

I. 225-228: again is this just based on the reduction of fire emissions or does this actually represent a change in the land carbon uptake due to changes in fire regimes?

The changes in population density since 1960 do lead to a truly enhanced land carbon uptake. We have now made this explicitly clear by adding another sentence.

I.243: and peatfires.

Done.

I. 255: The comparison with the observations however shows that the simple representation is able to capture the global trends in fire emissions.

Thank you. We do now make note that our simple approach works at the global scale.

I.273: What is the family of Canadian ESMs? any reference? Or delete this information, it does not seem important to know.

We have reworded the reference to family of Canadian ESMs?

I.289: typo: meteorological

Done.

I.264-265: I don't understand this. With your factorial simulations you can quantify how the land sink changes.

We did do our factorial simulations. In fact, the word "expected" in the sentence should not be there and that's what led reviewer #3 to think that we did not do our factorial simulations. We have reworded this sentence.

I. 475: ref 28 is not mentioned in the text.

Thank you. We have reformatted our references.

Figure 2: explain what the numbers in the graph are.

We have added additional text in the figure caption to make it easy to understand what the numbers in the bar plot imply.

Figure S6: you could add the GFAS dataset for carbon emission seasonality. It might also be interesting to add a trend analysis of the GFAS emissions to the paper, although the period is shorter for GFAS.

We unsuccessfully tried downloading the GFAS dataset (<http://apps.ecmwf.int/datasets/data/cams-gfas/>). However, the large size of these data made it difficult to download files and these data will, of course, require additional processing. While these data will provide an additional observation-based estimate to evaluate our model results these are not crucial to the conclusion drawn in our manuscript. We thank reviewer #2 to bring to our attention this data set which is not as well known as the GFED fire database. We will certainly download these data for future use in evaluating our model results.

Figure S7: Where does tau go to?

We have added additional text in the figure caption to explain how τ , the average fire duration, behaves as a function of fire extinguishing probability (q). The average fire duration τ is a function of fire extinguishing probability (q). In the absence of any human influence ($p_d=0$) $q=0.5$ and $\tau=1$ day. As population density increases, fire suppression increases which increases q and as a result decreases τ below 1.

Reviewers' comments:

Reviewer #1 (Remarks to the Author):

The authors' revision has clarified the issues I had with the original manuscript: the difference between the reduced burning caused by land use versus population. The manuscript now seems to me to be worthy of publication in Nature Communications.

One small point:

Reference #29 is missing the name of the journal.

Reviewer #3 (Remarks to the Author):

I find my comments well addressed in the revised version of the manuscript.

Reviewer #4 (Remarks to the Author):

This paper is disappointing for three reasons: firstly it really doesn't say anything particularly new, secondly it presents an entirely model-dependent analysis without showing that the model is fit-for-purpose or testing the results, and finally because the material is presented in such a way as to skate over key issues and uncertainties. I have only raised major issues of presentation and interpretation in this review, because until these are dealt with it does not seem worthwhile to comment on minor problems.

The methods section is inadequate to allow the reader to discover exactly what was done. The key to these findings lies in the model used. It is therefore not very helpful that there is no explicit description of the fire model in the methods. Although the basic equations are presented in Figure S8, there is no explanation or justification of these equations (for which the reader has to go back to a separate Melton and Arora paper). I think it is important that, in particular, the form of the relationships between population density and human ignitions and human suppression and the realism of these relationships are explored in this paper. I would suggest that the Methods section is

expanded to describe the model in sufficient detail to allow the reader to judge whether it is fit-for-purpose and whether the validity of the conclusions are compromised by the structure of the model.

I believe that a number of assumptions made in this model are highly questionable. The foremost example is the relationship between population density, human ignitions outside of croplands (which are not allowed to burn) and burnt area. The study by Bistinas et al. (2014, *Biogeosciences*) has shown that the unimodal relationship between population density and burnt area is an artifact of correlations with other controls on fire incidence and spread. This means that the use of such a relationship for modelling purposes is not supported by evidence; at best, results obtained using such a relationship may be correct for the wrong reasons. Knorr et al. (2014, *Biogeosciences*) show similar results to Bistinas et al., and this group have made simulations with “correct” treatments of the human effects on fire (see Knorr et al., 2016, *Nat. Clim Change* and Knorr et al., 2016, *Biogeosciences*) which seem to me to be a more profound analysis of fire regimes than offered in this paper.

Another modelling treatment that seems to be questionable is the supposed down-regulation of photosynthesis under nitrogen-limitation. It is fashionable to say this is the case, but I know of no evidence supporting this. In fact, models that include nitrogen-limitation seem to be unable to reproduce the basics of the interannual variation in CO₂ and its increase over the recent period (see e.g. Wenzel et al., 2014, *JGR*). Down-regulation is presented as a given in this paper, and there is no explanation of how it is achieved in the CLASS-CTM model, whether this behavior has been validated or what impact it has on the final results.

The recent Andela et al. paper in *Science* is cited in the text to support the idea that the increase in global crop area has led to a decrease in burned area (e.g. lines 70 and 120). No mention is made of one of the other conclusions of that paper – which is that state-of-the-art fire models from the FIREMIP intercomparison project are unable to reproduce recent trends in fire regimes. This despite the fact that the second author is a co-author on the Andela et al paper and that the CETM model was included in this comparison. I do not find it useful to analyse the impact of different “forcings” in a model which is basically unable to reproduce reality. This tells us about the model behavior, but not about the real world.

Comparisons with reality are obviously vital to support model-based inferences. The analyses presented here are not very convincing. While the general level of global area burnt is in the ballpark to satellite estimates, it is clear that the interannual variability in burnt area (Fig 1) nor the spatial patterns in emissions (Fig 2) are well-captured. The difference between simulated emissions and GFED emissions is perhaps more forgivable because the emissions are themselves model-based (CASA). However, given this discrepancy, I think that maps showing the spatial pattern of area burned should be included to allow the reader to determine whether the model is capturing more

than just the overall amount. I say this knowing full well, from the FIREMIP exercise, that the mapped patterns in burnt area are not realistic.

As a side note here: there are large differences between different burnt area products, as highlighted in a FIREMIP paper (Hantsen et al., 2016, Biogeosciences) on which the second author of this paper is a co-author. The inclusion of “small fires” in GFED4s is based on an algorithm rather than being pure observation, but was designed to capture agricultural fires – which this model set-up explicitly does not consider. So the authors really need to justify the use of a single product and of this single product in particular somewhere in this text.

I think some more effect is required to assess the reasonableness of the timing of the shift from positive to negative changes in burnt area and emissions (see e.g. lines 151-161). The argument here is that human ignitions contribute positively up to 1950 but that increased suppression means that by 1930 onwards there is a decline in burnt area and emissions. Comparison with the charcoal data is not satisfactory – the charcoal plot shown on Fig 1 implies relatively stable conditions up to 1920 and a decline thereafter (although the authors persist on stating that it shows a decline after 1930 congruent with the model results, it categorically does not); the model appears to show an increase in the first part of the record. The latest version of the charcoal record (2016), which has double the number of sites and better spatial coverage, is not used here for comparison (although shown in the SI) because it shows an upturn over the last 5 years of the record and because it does not compare as well with the simulations (lines 213-214). Oh dear. If the authors do not think that that charcoal data is a reliable source of information, then why not use some other source of information about historical changes in fire regimes? How do these trends compare with the Mouillot and Field (2005) 20th century reconstructions for example?

I have no idea what “fire activity” is (see e.g. lines 195-198, caption to figure 1). Does this mean fire frequency, fire intensity, total number of fires, burnt area, biomass consumed. The use of such vague phrases is not helpful in science because it basically means that anything goes. The amount of charcoal produced in a fire is related to the amount of biomass consumed, so if charcoal is used as a proxy for fire it is a proxy for biomass burned.

Sandy P. Harrison

16 September 2017

Response to second round of reviewer comments for “Reduction in global area burned and wildfire emissions since 1930s enhances carbon uptake by land”

This paper is disappointing for three reasons: firstly it really doesn't say anything particularly new, secondly it presents an entirely model-dependent analysis without showing that the model is fit-for-purpose or testing the results, and finally because the material is presented in such a way as to skate over key issues and uncertainties. I have only raised major issues of presentation and interpretation in this review, because until these are dealt with it does not seem worthwhile to comment on minor problems.

1) We are not aware of any publication that quantifies the effect of decreasing fire emissions on land carbon sink so the results in the manuscript are indeed new. The effect of decreasing fire emissions on land carbon sink cannot be diagnosed without a model-based analysis since reduced fire emissions from increasing crop area do not yield a carbon sink as we extensively discuss in our manuscript. We are not aware of any such model based analysis which explicitly tries to tease out the effect of recent reduced burning on the land carbon sink.

2) The version of the manuscript reviewed by Dr. Harrison assessed the model against five separate measures to illustrate that the model does behave reasonably realistically to address the scientific question of the role of decreasing fire emissions on land carbon sink. These included the 1) zonal distribution of area burned, 2) trends in global area burned and fire CO₂ emissions, 3) geographical distribution of trends in area burned, 4) simulated net atmosphere-land CO₂ flux, and 5) seasonality of simulated global fire CO₂ emissions – all of which compare reasonably with observation-based estimates.

In addition we now also show that 1) simulated geographical distribution of area burned compares reasonably with observation-based estimates and 2) also report the correlation coefficient between simulated and GFED4.1s annual area burned estimates for the 1997-2014 period ($R^2=0.75$).

3) We now provide additional evidence (as discussed below) to strengthen our argument that the modelled response to increase in population density is fairly realistic. We believe this is the key uncertainty that Dr. Harrison is referring to.

The methods section is inadequate to allow the reader to discover exactly what was done. The key to these findings lies in the model used. It is therefore not very helpful that there is no explicit description of the fire model in the methods. Although the basic equations are presented in Figure S8, there is no explanation or justification of these equations (for which the reader has to go back to a separate Melton and Arora paper). I think it is important that, in

particular, the form of the relationships between population density and human ignitions and human suppression and the realism of these relationships are explored in this paper. I would suggest that the Methods section is expanded to describe the model in sufficient detail to allow the reader to judge whether it is fit-for-purpose and whether the validity of the conclusions are compromised by the structure of the model.

We have now included additional details in the Methods section with a particular focus on the role of population density in affecting human caused fire ignitions and fire suppression.

I believe that a number of assumptions made in this model are highly questionable. The foremost example is the relationship between population density, human ignitions outside of croplands (which are not allowed to burn) and burnt area. The study by Bistinas et al. (2014, Biogeosciences) has shown that the unimodal relationship between population density and burnt area is an artifact of correlations with other controls on fire incidence and spread. This means that the use of such a relationship for modelling purposes is not supported by evidence; at best, results obtained using such a relationship may be correct for the wrong reasons. Knorr et al. (2014, Biogeosciences) show similar results to Bistinas et al., and this group have made simulations with “correct” treatments of the human effects on fire (see Knorr et al., 2016, Nat. Clim Change and Knorr et al., 2016, Biogeosciences) which seem to me to be a more profound analysis of fire regimes than offered in this paper.

This appears to be the primary concern of Dr. Harrison and is spurred by Figure S8 of the previous version of our manuscript together with Figure 1 (the panel showing effect of population density on area burned after log transformed multiple linear regression, column 2, row 3) and Figure 4 (panel showing burnt area fraction versus log of population density, column 2, row 1) from Bistinas et al. (2014) (on which Dr. Harrison is the second author). In our Figure S8, Dr. Harrison found that model equations yield a relationship between area burned and population density that is unimodal (i.e. area burned peaks at some population density) and inconsistent with panel in column 2 and row 3 of Figure 1 of Bistinas et al. (2014) which suggests that area burned always decreases with an increase in population density. A subtle point that is associated with Figure S8 is that the model relationship was shown for zero lightning. As soon as probability of fire due to lightning ($P_{i,n}$) increases above zero then indeed the primary role of increase in population density is to reduce area burned, although the functional form of our relationship is different than Bistinas et al. (2014) and Knorr et al. (2016). We have now included three additional figure panels in the revised manuscript (see Figure R1 below) to show this behaviour.

Figure R1

We also plotted a graphic similar to Figure 4 of Bistinas et al. (2014) to compare the emergent behavior between area burned and population density for CLASS-CTEM and GFED v 4.1s satellite-based product and this is shown below in panels a) and b) of Figure R2. The color scale indicates the density of points.

Panel c) of Figure R2 shows the same results as in panels a) and b) but both area burned and population density are plotted on a log scale, and the model and GFED4.1s based results are on the same plot. These figures show that the emergent relationships between area burned and population density are very similar in the model and GFED4.1s product. That the model reproduces the observation-based emergent behaviour in this regard provides confidence in the model results. In addition, both model and GFED4.1s based area burned peak at around 20 people/km². Some differences remain between this emergent behaviour for model and GFED4.1s estimates but Figure R2 confirms that the emergent behaviour between area burned and population density in the model is realistic. We have included panel (c) of the above figure in our revised manuscript.

Figure R2

Dr. Harrison made the case that based on Figure 1 of Bistinas et al. (2014) an increase in population density always decreases area burned. Although our model does incorporate this broad behaviour (except for the case when lightning is zero, as mentioned above) we still want to refute this based on following five reasons.

1. First, it seems unreasonable to assume that extremely low population densities in remote areas people have a higher potential to suppress fires than to ignite fires. Clearly a single person living in 10 km^2 area (density = 0.1 person/km^2) can light up a fire more easily than suppress an ongoing lightning-caused fire. Both landscape fragmentation and the ability to suppress fires through fire-fighting efforts require economic abilities and social infrastructure that are typically not possible at extremely low population densities.
2. Second, while Dr. Harrison used the panel for population density (column 2, row 3) from Figure 1 of Bistinas et al. (2014) to make her case (that increasing population densities always decreases burning), if we were to see the panel for pasture area (column 4, row 1) from Figure 1 of Bistinas et al. (2014) it shows that

burning increases as pasture area increases. An increase in global pasture area (along with the increase in crop area) is clearly the result of increase in global population. Since humans increase pasture area, and area burned increases with an increase in pasture area (likely due to fires caused by humans), the panel for pasture area (column 4, row 1) from Figure 1 of Bistinas et al. (2014) implies that an increase in population does increase area burned in pasture areas.

3. Third, there is a large spread in the panel for population density in Figure 1 of Bistinas et al. (2014) at low population densities so a linear relationship (on log scale) is probably not the best fit at low population densities.
4. Fourth, there is ample literature that points to cultural, behavioural and socio-economic reasons behind why humans light fires. For example, Stocker and Mott (1981), report that in Northern Australia fires ignited by lightning strikes are few in number, limited to a small proportion of total area, and restricted to a period between 1 and 2 months at the start of the wet season. Since there is little lightning during the dry season, the dry season fires in Northern Australia are set by humans for purposes associated principally with indigenous land management practices, the pastoral industry, and conservation management (Russell-Smith et al., 1997). A 2002 British Broadcasting Corporation (BBC) article (<http://news.bbc.co.uk/1/hi/sci/tech/1977986.stm>, May 2002) entitled “Deliberate fires set Africa ablaze” discusses reasons which include the use of fire to clear vegetation and create better grazing land, the actions of honey gatherers who light fires to smoke bees out of trees, and arsonists who hope to acquire land after its burned. With these reasons and at low population densities where landscape fragmentation is not enough and the ability to suppress fires through fire-fighting efforts is not fully developed area burned can increase with population density.
5. Finally, the relationship between area burned and population density implemented by Knorr et al. (2016) which Dr. Harrison describes as “correct” is based on the following equation from Knorr et al. (2016)

$$A(t) = a(i) F^b N_{\max}(t)^c \exp(-ep) \quad (1)$$

In equation (1), $A(t)$ is the area burned for year t , $a(i)$ is a constant for biome i , F is the fraction of photosynthetically absorbed radiation (a proxy for available vegetation and above ground biomass), N_{\max} is the Nesterov index (a measure of climate which combines temperature and precipitation), and finally area burned continuously decreases as population density, p , increases based on $\exp(-ep)$. a , b , c and e are fitted constants. Equation (1) implies that when population density is zero, then $\exp(-e.0) = 1$ and area burned will be maximum, regardless of natural ignition sources, if there is enough vegetation biomass and climate is conducive to fire. The model essentially makes the unrealistic assumption that natural

ignition sources are unlimited and not a constraint on burning. This model is applied at an annual time step.

We are unable to comment in detail on the Knorr et al. (2016) approach because it is not our model but it appears that the approach (with human always and only causing fire suppression) likely works because it is applied at an annual time step and because it unrealistically assumes ignition sources are unlimited (which also means human-caused fire ignitions are lumped together with natural fire ignitions). It follows then the only possible role left for humans is to suppress fire.

We cannot incorporate such an approach in our modelling framework. Our approach uses both lightning and human caused ignitions separately to determine probability of fire conditioned on availability of an ignition source, and also treats fire-ignition and suppression caused by humans separately. In addition, the area burned in the CLASS-CTEM model is determined at a daily time step (not annually).

The above discussion suggests that the assumption that the net effect of humans is always and to only cause fire suppression, and the unrealistic assumption of unlimited natural ignitions, possibly yields correct results for the wrong reasons.

Another modelling treatment that seems to be questionable is the supposed down-regulation of photosynthesis under nitrogen-limitation. It is fashionable to say this is the case, but I know of no evidence supporting this. In fact, models that include nitrogen-limitation seem to be unable to reproduce the basics of the interannual variation in CO₂ and its increase over the recent period (see e.g. Wenzel et al., 2014, JGR). Down-regulation is presented as a given in this paper, and there is no explanation of how it is achieved in the CLASS-CTM model, whether this behavior has been validated or what impact it has on the final results.

The information about down-regulation of photosynthesis was included in the manuscript in response to comments from first round of reviews – i.e., if the model contains coupling of terrestrial N and C cycles. The model includes down-regulation of photosynthesis to emulate nitrogen constraint and is based on response of plants grown in ambient and elevated CO₂ environment as inferred from meta-analyses of such studies. The overall objective of this exercise is to obtain reasonable rate of increase of gross primary productivity over the historical period and the approach and its rationale and evaluation are already published (Arora and Scinocca, 2016; Arora et al., 2009). Regardless, this aspect has minimal effect on the behaviour of fire over the historical period and not directly relevant to fire behaviour. This is (was) illustrated by the factorial simulations which (were) are summarized in Figure 4 (3) of the revised (original) manuscript which analyzes the effect of different forcings. This analysis of different

forcings was found not useful by Dr. Harrison (see her comments below) and yet it is this analysis which allows us to confirm that over the historical period increasing CO₂ doesn't affect area burned considerably. This analysis also confirms that how the model treats photosynthesis down-regulation is irrelevant for the behaviour of area burned and fire emissions over the historical period.

The recent Andela et al. paper in Science is cited in the text to support the idea that the increase in global crop area has led to a decrease in burned area (e.g. lines 70 and 120). No mention is made of one of the other conclusions of that paper – which is that state-of-the-art fire models from the FIREMIP intercomparison project are unable to reproduce recent trends in fire regimes. This despite the fact that the second author is a co-author on the Andela et al paper and that the CETM model was included in this comparison. I do not find it useful to analyse the impact of different “forcings” in a model which is basically unable to reproduce reality. This tells us about the model behavior, but not about the real world.

We did not report on results from other models used in Andela et al., (2017) but now we do. We now explicitly note in the Methods section that not all models reproduce the recent decreasing trend in global area burned in Andela et al., (2017). Five of the participating models (including CLASS-CTEM) simulated a decrease in global area burned over the 1997-2014 period while the remaining four simulated an increase over the same period. Of course, we cannot be expected to comment on the reasons for the trends of other models. It also does not imply either that if other models do not simulate the observed negative trend in area burned over the historical period then CLASS-CTEM's behaviour is unrealistic.

In regards to the analyses of impact of different forcings, please note that the effect of decreasing fire emissions on land carbon sink cannot be diagnosed without a model-based analysis since reduced fire emissions from increasing crop area do not yield a carbon sink as we extensively discuss in our manuscript. The factorial experiments we have performed with our model by turning one driver on at a time allows us to quantify the effect of reducing fire emissions on land carbon sink and interpret the results of the historical simulation with all forcings. These experiments also allowed us to conclude that over the historical period increasing CO₂ and changing climate have minimal effect on fire behaviour (as mentioned above in the context of photosynthesis downregulation) similar to Knorr et al. (2016) who also performed simulations with different forcings.

Comparisons with reality are obviously vital to support model-based inferences. The analyses presented here are not very convincing. While the general level of global area burnt is in the ball-park to satellite estimates, it is clear that the interannual variability in burnt area (Fig 1) **nor the spatial patterns in emissions (Fig 2) are well-captured.** The difference between simulated

emissions and GFED emissions is perhaps more forgivable because the emissions are themselves model-based (CASA). However, given this discrepancy, I think that maps showing the spatial pattern of area burned should be included to allow the reader to determine whether the model is capturing more than just the overall amount. I say this knowing full well, from the FIREMIP exercise, that the mapped patterns in burnt area are not realistic.

As mentioned earlier, we compared the performance of CLASS-CTEM fire behaviour against five measures in the version of the manuscript that Dr. Harrison reviewed and the model results compare reasonably with observation-based estimates for the present-day.

Dr. Harrison noted that “it is clear that the interannual variability in burnt area (Fig 1) nor the spatial patterns in emissions (Fig 2) are well-captured”. In fact, Figure 2 showed the geographical distribution of trend in area burned and not the geographical distribution of area burned. The geographical distribution of the trend in area burned is even harder to simulate than area burned itself because the trends depend on the response of model to changes in population density and increase in crop area – yet the model does a reasonable job in reproducing the spatial trend in emissions (except in the boreal region). The revised manuscript now includes an additional figure in supplementary information that shows geographical distribution of simulated area burned compares reasonably with the two satellite-based estimates.

In regards to inter-annual variability in annual burned area, in fact the correlation coefficient between simulated and GFED4.1s annual area burned estimates is 0.75 for the 1997-2014 period. We now mention this in the revised manuscript.

As a side note here: there are large differences between different burnt area products, as highlighted in a FIREMIP paper (Hantsen et al., 2016, Biogeosciences) on which the second author of this paper is a co-author. The inclusion of “small fires” in GFED4s is based on an algorithm rather than being pure observation, but was designed to capture agricultural fires – which this model set-up explicitly does not consider. So the authors really need to justify the use of a single product and of this single product in particular somewhere in this text.

In reading of Randerson et al. (2012) it is not clear to us if the small fires contribution to GFED was designed specifically for agricultural fires. Randerson et al. (2012) note that in addition to agricultural settings, small fires occur where the use of prescribed fires is important for ecosystem management, where wildland fires are suppressed near human settlements, in ecosystems with heterogeneous patches of land cover that limit the continuity of dry fuels and in places and times where fuel moisture and atmospheric conditions do not sustain high fire spread rates. In addition, they note small fires occur not only in croplands but also in wooded savanna and tropical forest biomes.

Regardless, we now compare the time series of simulated annual area burned with several proxy and area burned estimates – these include both charcoal indices (released in 2008 and 2016), Mouillot and Field reconstruction, and the European Space Agency’s Climate Change Initiative (ESA CCI) and GFED4.1s area burned products all on one plot.

I think some more effect is required to assess the reasonableness of the timing of the shift from positive to negative changes in burnt area and emissions (see e.g. lines 151-161). The argument here is that human ignitions contribute positively up to 1950 but that increased suppression means that by 1930 onwards there is a decline in burnt area and emissions. Comparison with the charcoal data is not satisfactory – the charcoal plot shown on Fig 1 implies relatively stable conditions up to 1920 and a decline thereafter (although the authors persist on stating that it shows a decline after 1930 congruent with the model results, it categorically does not); the model appears to show an increase in the first part of the record. The latest version of the charcoal record (2016), which has double the number of sites and better spatial coverage, is not used here for comparison (although shown in the SI) because it shows an upturn over the last 5 years of the record and because it does not compare as well with the simulations (lines 213-214). Oh dear. If the authors do not think that that charcoal data is a reliable source of information, then why not use some other source of information about historical changes in fire regimes? How do these trends compare with the Mouillot and Field (2005) 20th century reconstructions for example?

The reason we included the figure with 2016 charcoal record in supplementary information was that the increase in charcoal index for the period centered on 2010 is not consistent with satellite-based area burned estimates which show a continuing decreasing trend in area burned. Charcoal indices are proxies with large and unquantified uncertainties.

We have now reworded our text to differentiate what the model does and what the charcoal indices show. As mentioned above, we now compare all observation-based and proxy estimates of burned area (Mouillot and Field, and ESA CCI and GFED4.1s area burned products) and charcoal indices - all on one plot (Figure 2 in the revised manuscript). In the end, the charcoal indices and Mouillot and Field reconstruction are proxies and satellite-based area burned estimates provide the best available means to assess model behaviour. In this regard, model results provide means to look back in time - provided, of course, model performance for the present day and its response to primary forcings is reasonably realistic – which we believe we have shown to be the case.

I have no idea what “fire activity” is (see e.g. lines 195-198, caption to figure 1). Does this mean fire frequency, fire intensity, total number of fires, burnt area, biomass consumed. The use of such vague phrases is not helpful in science because it basically means that anything goes. The amount of charcoal produced in a fire is related to the amount of biomass consumed, so if charcoal is used as a proxy for fire it is a proxy for biomass burned.

We have replaced “fire activity” with “burning” following Marlon et al. (2012) use of the term to describe charcoal indices.

References

- Andela, N., Morton, D.C., Giglio, L., Chen, Y., van der Werf, G.R., Kasibhatla, P.S., DeFries, R.S., Collatz, G.J., Hantson, S., Kloster, S., et al. (2017). A human-driven decline in global burned area. *Science* 356, 1356.
- Arora, V.K., and Scinocca, J.F. (2016). Constraining the strength of the terrestrial CO₂ fertilization effect in the Canadian Earth system model version 4.2 (CanESM4.2). *Geosci. Model Dev.* 9, 2357–2376.
- Arora, V.K., Boer, G.J., Christian, J.R., Curry, C.L., Denman, K.L., Zahariev, K., Flato, G.M., Scinocca, J.F., Merryfield, W.J., and Lee, W.G. (2009). The Effect of Terrestrial Photosynthesis Down Regulation on the Twentieth-Century Carbon Budget Simulated with the CCCma Earth System Model. *J. Clim.* 22, 6066–6088.
- Bistinas, I., Harrison, S.P., Prentice, I.C., and Pereira, J.M.C. (2014). Causal relationships versus emergent patterns in the global controls of fire frequency. *Biogeosciences* 11, 5087–5101.
- Knorr, W., Arneth, A., and Jiang, L. (2016). Demographic controls of future global fire risk. *Nat. Clim. Change* 6, 781.
- Marlon, J.R., Bartlein, P.J., Gavin, D.G., Long, C.J., Anderson, R.S., Briles, C.E., Brown, K.J., Colombaroli, D., Hallett, D.J., Power, M.J., et al. (2012). Long-term perspective on wildfires in the western USA. *Proc. Natl. Acad. Sci.* 109, E535–E543.
- Randerson, J.T., Chen, Y., van der Werf, G.R., Rogers, B.M., and Morton, D.C. (2012). Global burned area and biomass burning emissions from small fires. *J. Geophys. Res. Biogeosciences* 117, n/a-n/a.
- Russell-Smith, J., Ryan, P.G., and Durieu, R. (1997). A LANDSAT MSS-Derived Fire History of Kakadu National Park, Monsoonal Northern Australia, 1980-94: Seasonal Extent, Frequency and Patchiness. *J. Appl. Ecol.* 34, 748–766.
- Stocker, G. C., and J. J. Mott (1981), Fire in the tropical forests and woodlands of northern Australia, in *Fire and the Australian Biota*, edited by A. M. Gill, R. H. Groves, and I. R. Noble, pp. 425–439, Aust. Acad. of Sci., Canberra, ACT, Australia.

REVIEWERS' COMMENTS:

Reviewer #3 (Remarks to the Author):

I am commenting here on the review of Dr. Harrison and the author's reply.

The main concerns of the review are that it doesn't say anything new, it is entirely dependent on the model without testing the results, and that uncertainties are not mentioned.

I agree with the authors that estimating the effect of decreasing fire emissions on the carbon cycle is novel and interesting. It increases our understanding of trends in the global carbon sink and shows that changes in fire regimes can significantly alter this sink. Often the relation between climate change and fire emissions is assumed to be a positive feedback based on plausibility. It is a major advancement to face such assumptions with numerical model simulations

I also think that the authors use available observations appropriately to evaluate their model and the model behaviour looks reasonably well to support the overall conclusions of the manuscript. Global models most of the time rather can represent large scale patterns, however, also the conclusions and results presented are not based on small scale variations of burned area. The authors extend their evaluation by including additional datasets. I am not aware of additional datasets that could provide better constraints or support the conclusions further. Lack of observational evidence is in the end one of the reasons we are using models to understand the past.

The discussion of uncertainty has strongly improved in the revised manuscript. In my opinion the analysis showing that including lightning changes the relationship between humans and burned area is a major step forward. It not only improves the discussion of uncertainties but it also helps to solve the apparent contradiction between increases in ignitions due to human activity and the results of studies investigating the relationship between humans and burned area based on satellite data which find that the suppression dominates. I would also like to support their point 3 on page 5, saying that a linear relationship (on log scale) is not a good model. Moreover a relationship with a maximum as used in the model here cannot be excluded by the definition of a linear model. Additional support in literature on the increasing effects of humans on fire occurrence can be found in literature on charcoal records (McWethy et al. 2010). The point that natural ignitions can be limiting for burned area was recently supported by a remote sensing study where they find that years with high fire occurrence are years with increased lightning rates and that the high fire occurrence is due to a higher number of fires not a higher size of fires (Veraverbeke et al. 2017).

The review also criticises that the down-regulation of photosynthesis due to nitrogen is included. To my knowledge nitrogen limitation is an important and uncertain factor, showing that this does not considerably influence the results does give more confidence in the study.

The criticism that global fire models don't reproduce the trend does indeed not apply for the CLASS-CTEM model which does model a decreasing trend for the time period covered by satellite data.

In summary I do not share the concerns that the model does not have enough support based on the model evaluation and I think the criticism about the model structure is well addressed in the reply as well as in the revised manuscript.

Veraverbeke, S., Rogers, B. M., Goulden, M. L., Jandt, R. R., Miller, C. E., Wiggins, E. B. and Randerson, J. T.: Lightning as a major driver of recent large fire years in North American boreal forests, *Nat. Clim. Chang.*, 7(7), 529–534, doi:10.1038/nclimate3329, 2017.

McWethy, D. B., Whitlock, C., Wilmschurst, J. M., McGlone, M. S., Fromont, M., Li, X., Dieffenbacher-Krall, A., Hobbs, W. O., Fritz, S. C. and Cook, E. R.: Rapid landscape transformation in South Island, New Zealand, following initial Polynesian settlement, *Proc. Natl. Acad. Sci.*, 107(50), 21343–21348, doi:10.1073/pnas.1011801107, 2010.